# Analysis of the current status and influencing factors of kinesiophobia in tumor patients with peripherally inserted central catheter: A cross-sectional study

Xiaohua Zhu[1☯], Yan Wu[2☯], Rong Li [2¤a*], Xiaozhu Qiao[3*], Ying Yang[2‡], Fang Chen[2‡]

**1** Vascular Access Clinic, Xinghua people's Hospital Affiliated to Yangzhou University, Xinghua, Jiangsu, China, **2** Vascular Access Clinic, Taizhou Third People's Hospital, Taizhou, Jiangsu, China, **3** Nursing Department, Taizhou People's Hospital, Taizhou, Jiangsu, China

☯ These authors contributed equally to this work.
‡ These authors contributed equally to this work.
¤a Vascular access clinic, Taizhou Third People's Hospital, No.98 Chungang East Road, Gaogang District, Taizhou City, Jiangsu Province, China
* lirong19900916@163.com (RL); 18936798035@163.com (XQ)

## Abstract

### Background

Peripherally Inserted Central Catheters (PICCs) are widely utilized in tumor patients due to their lower risk of complications, extended indwelling duration, reduced local tissue trauma, and overall cost-effectiveness. Based on the Health Action Process Approach (HAPA) theory, this study aims to explore the current status and influencing factors of kinesiophobia in tumor patients with PICCs. The study provides reference for clarifying the mechanism of kinesiophobia and developing nursing intervention plans.

### Method

Through convenience sampling, 162 tumor patients who underwent PICC maintenance in three hospitals in Jiangsu Province from December 4th, 2023, to December 31st, 2024 were selected. The patient general information questionnaire, Tampa scale of kinesiophobia, medical coping modes questionnaire, exercise self-efficacy scale, risk perception questionnaire, outcome expectation scale, exercise intention scale, and social support rating scale were used for evaluation.

### Result

Tumor patients carrying PICC had a kinesiophobia score of $20.11 \pm 6.94$ points, and 42.59% of tumor patients with PICC had kinesiophobia. The results of multiple linear regression showed that the duration of catheter placement($t = -3.506, P = 0.001$),

**Data availability statement:** The data files are available in the Science Data Bank(Science DB): https://www.scidb.cn/anonymous/YXIISVpu. (The dataset will be made accessible upon reasonable request. The corresponding accession number/DOI will be provided during the production stage if the manuscript is accepted for publication.).

**Funding:** This work was supported by the 2024 Taizhou "Fengcheng Yingcai Program" Young Science and Technology Talent Support Project (to R.L.) and the Jiangsu Provincial Young Science and Technology Talent Support Project (Grant No. JSTJ-2024-624) (to R.L.).

**Competing interests:** All authors declare no conflicts of interest.

pain(t = 2.652,P = 0.009), exercise self-efficacy(t = −3.891,P < 0.001), and risk perception(t = 3.157,P = 0.002) are the main influencing factors of kinesiophobia in tumor patients with PICC.

## Conclusion

The findings underscore a significant clinical concern regarding kinesiophobia among tumor patients with PICC. It is essential for nursing staff to implement systematic assessments and tailored interventions aimed at mitigating kinesiophobia. Addressing this issue can contribute to reducing associated adverse reactions and improving patient mobility and overall quality of life.

## Introduction

Malignant tumors represent a major public health challenge in China, with incidence and mortality rates ranking among the highest globally [1]. Both domestic and international guidelines recognize that exercise testing and intervention are generally safe and effective for tumor patients. Proper exercise and physical activity can affect the tumor microenvironment, inhibit cancer cells, enhance immunity, and delay tumor progression of cancer patients [2]. However, recent meta-analyses have shown that 47% of tumor patients experience cancer-related pain [3], which can directly contribute to kinesiophobia—a debilitating fear of movement or physical activity due to pain or injury concerns [4]. For patients requiring chemotherapy, the peripherally inserted central catheter (PICC) has become the most widely used central venous catheter in China due to its minimally invasive nature, avoidance of repeated punctures, and convenience for safe medication [5,6]. During PICC catheterization, appropriate physical activity is recommended to lower the risk of catheter-associated deep vein thrombosis (CA-DVT) [7]. Nonetheless, some patients may avoid such activities due to concerns about catheter dislodgement, fracture, or limitations in performing movements with moderate amplitude or frequency [8–10]. Compared to other cancer populations, those with a PICC may exhibit significantly greater kinesiophobia, which in turn increases thrombosis risk and impairs daily functioning and independence.

Exercise phobia, also known as kinesiophobia, refers to an irrational fear behavior towards exercise and physical activities that manifests in physiological symptoms, psychological discomfort, or fatigue in response to external stimuli [11,12]. While kinesiophobia has been investigated in various clinical populations—such as patients with coronary heart disease [13], migraines [14], and orthopedic postoperative patients [15], research focusing on cancer patients with PICC remains limited.. Previous studies on tumor patients with PICC have mostly focused on catheter-related complications [16,17], with less attention paid to the side effects caused by reduced exercise.

Previous theoretical models related to kinesiophobia often used fear avoidance models, which centered around pain [18]. Nowadays, with the expansion of research subjects on kinesiophobia, the existence of fear of exercise suggests not only

avoidance of pain, but also other reasons. The Health Action Process Approach (HAPA) model was selected because it directly addresses the dynamic nature of health behaviors like kinesiophobia. Its phased structure, which encompasses both motivational and volitional phases, is ideal for investigating how such fear develops and can be mitigated over time. Given the detrimental impact of kinesiophobia on health behavior planning and execution in PICC-carrying cancer patients, this study adopts the HAPA model to analyze its influencing factors and underlying mechanisms. The HAPA model integrates stage-specific and continuous behavioral processes, and includes key constructs such as self-efficacy, risk perception, outcome expectancies, intention, planning, and action, making it a suitable theoretical framework for this investigation.

Therefore, this cross-sectional study aims to observe tumor patients with PICC, identify factors associated with kinesiophobia, and provide a basis for clarifying its mechanisms and developing targeted nursing interventions.

## Subject and method

### Survey subjects

Convenience sampling was used to select tumor patients who underwent PICC catheter maintenance at three hospitals in Taizhou City, from December 4th, 2023, to December 31st, 2024, as the research subjects. There are a total of 27 variables analyzed in this survey, and the Kendall method [19] is used to take 5 times the number of variables as the required sample size, which is 135 cases. Considering the possibility of missing sample size, an additional 20% of the sample will be added to ultimately include 162 cases. Inclusion criteria: ①Patients diagnosed with malignant tumors by the attending physician; ②Age ≥ 18 years old; ③First insertion of PICC, with a catheterization time of ≥ 2 weeks; ④Can independently complete the questionnaire or complete the questionnaire with the assistance of the researcher. Exclusion criteria: ①Have absolute contraindications to physical activity; ②Have severe comorbid conditions that could independently limit survival or confound the assessment; ③Have a documented history of mental illness or cognitive impairment; ④Have severe neurological or musculoskeletal disorders that substantially limit voluntary limb movement. The detailed selection process is presented in (Fig 1).

### Investigation tools

**General information questionnaire.** Based on literature review, the group discussed the design of the questionnaire. The questionnaire includes age, gender, education level, monthly household income, place of residence, time of catheter insertion, PICC placement location (left arm, right arm or other), occurrence of complications (occurrence time), presence of foreign body sensation, pain, and number of intubation attempts.

**Tampa scale of kinesiophobia.** The Tampa Scale of Kinesiophobia (TSK-11) was used to evaluate tumor patients carrying PICC catheters. This scale was revised and simplified by Woby [20] in 2005 based on the original Tampa Scale of Kinesiophobia. The revised scale consists of 11 items, divided into three dimensions: activity attitude, activity behavior cognition, and activity behavior. For each item, survey respondents are required to rate their level of agreement, with 1 point for strongly disagree, 2 points for disagree, 3 points for agree, and 4 points for strongly agree. The total score ranges from 11 to 44 points, with higher scores indicating a higher degree of kinesiophobia. The Chinese version of the scale has good internal consistency and test-retest reliability, with a Cronbach'α coefficient of 0.883 and an within-group correlation coefficient (ICC) of 0.798 [21]. Jimenez et al. [22] classified fear levels into four levels: no fear (≤ 17 points), mild fear (18–24 points), moderate fear (25–31 points), severe fear (32–38 points), and extreme fear (39–44 points).

**Medical coping modes questionnaire.** The Medical Coping Modes Questionnaire (MCMQ) revised by Shen Xiaohong [23] was used to evaluate patients' coping modes in various medical situations. The scale consists of 20 items, divided into three dimensions: yielding, avoidance, and facing. The highest score is obtained in one dimension, indicating that patients tend to prefer the coping mode of that dimension.

## Participant inclusion and exclusion process diagram

**Fig 1. Participant inclusion and exclusion process diagram.**

**Exercise self-efficacy scale.** The Exercise Self-Efficacy Scale (ESE) designed by Bandura and localized by Tung et al. [24] was used to evaluate tumor patients with PICC. This scale has been widely used in the assessment of exercise self-efficacy among chronic disease patients abroad. This scale contains a total of 18 items, with each item ranging from 0 to 100 points. A score of 0 indicates complete inability, 50 indicates moderate certainty, and 100 indicates complete certainty. The average score of the 18 items is used as the scale score, and the higher the score, the higher the level of exercise self-efficacy.

**Risk perception questionnaire.** The survey was conducted using the medical risk perception questionnaire developed by Fang Lei et al [25]. This scale consists of 12 items, divided into three dimensions: economic risk, physical diagnosis and treatment risk, and social psychological risk. It uses the Likert 1–5 scoring method, with 1 point indicating very worried and 5 points indicating not worried at all. The higher the total score of each item, the higher the risk perception.

**Exercise outcome expectation questionnaire.** The survey was conducted using the exercise outcome expectation scale developed by Renner et al. [26]. This scale includes ten positive outcome expectations and three negative outcome expectations, using the Likert 1–5 scoring method, with 1 point indicating complete impossibility and 5 points indicating complete possibility. Negative outcomes are scored in reverse, and the total score of all items is the outcome expectation total score. The higher the total score, the higher the positive expectation.

**Exercise intention.** The survey was conducted using the exercise intention scale developed by Duan. [27] The scale consists of four items, and the survey participants answer the possibilities described in each item using the Likert 1–5 scoring method. One point is complete impossibility, and five points are complete possibility. The exercise intention is determined by the total score of each item, and the higher the score, the higher the exercise intention.

**Social support rating scale.** The Social Support Rating Scale (SSRS) developed by Xiao Shuiyuan et al. [28] was used for evaluation. This scale consists of 10 items, divided into three dimensions: objective support, subjective support, and support utilization. It uses a Likert 1–4 point scoring method, with 1 point indicating strongly disagree and 4 points indicating strongly agree. The higher the total score, the higher the level of social support.

## Data collection and quality control

Two nurses who regularly work in the vascular access clinic were trained before the investigation. After passing the training assessment, the two nurses took turns collecting data. During the survey, a unified guiding language was used to inform patients of the research purpose and questionnaire filling method, and the questionnaire was distributed after obtaining informed consent. During the filling process, no hints was given. If patients need assistance with questionnaire filling, the investigator explained it through unified explanatory words. After completing the questionnaire, it was collected immediately. One researcher transcribed it into electronic data within 48 hours, and the other verified it.

## Statistical methods

SPSS 26.0 was used for data processing. Pearson correlation analysis was used between kinesiophobia and various variables. For univariate analysis, continuous variables were compared using independent samples t-tests, and categorical variables were compared using Chi-square tests or one-way Analysis of Variance (ANOVA), as appropriate. Variables yielding a P-value $< 0.05$ in these initial omnibus tests were considered potential candidates for inclusion in the subsequent multivariate model. We did not perform post-hoc pairwise comparisons for multi-category variables; therefore, a Bonferroni correction was not applied. The focus of the univariate analysis was on screening variables for the regression model, not on establishing specific between-group differences. Multiple linear regression analysis was used for multivariate analysis. A multiple linear regression model was employed to identify the factors associated with kinesiophobia scores. Prior to the final analysis, the underlying assumptions of linear regression were thoroughly examined. The linearity between continuous independent variables and the dependent variable was assessed visually using partial regression plots and was found to be satisfactory. The independence of residuals was confirmed by a Durbin-Watson statistic close to 2. The normality of the residuals was verified using a histogram and a normal Q-Q plot. Homoscedasticity (homogeneity of variance) was evaluated by plotting the standardized residuals against the standardized predicted values, which revealed no evident pattern. Finally, the absence of multicollinearity was ensured, as all variance inflation factor (VIF) values were below 10.

## Ethical statement

This study was approved by the ethics committee of Taizhou Third People's Hospital, approval No.TZSRY-LS-2023YL-026. Before the questionnaire was administered, each participant provided written informed consent. This study follows the guidelines set out in the declaration of Helsinki.

## Results

### Questionnaire results on kinesiophobia, medical coping modes, and exercise self-efficacy from tumor patients with PICC catheterization

This study collected a total of 162 cases, and no patients withdrew midway, it may be attributed to three key factors. First, the fixed weekly schedule of PICC maintenance allowed patients ample and flexible opportunities to complete the survey during routine clinic visits. Second, researchers provided thorough communication and used patient-friendly language during the informed consent process, enabling participants to make well-informed decisions. Finally, the established trust and familiarity between patients and the consistent clinic staff created a supportive environment that encouraged ongoing engagement. Table 1 presents the baseline characteristics of the study population. The patient's kinesiophobia score was $20.11 \pm 6.94$ points, and the incidence of kinesiophobia was 42.59% (69 cases). Among them, activity attitude was $4.82 \pm 2.56$ points, activity cognition was $10.67 \pm 2.94$ points, and activity behavior wad $5.20 \pm 1.69$ points.

Among the research participants, 54 cases (33.33%) tended to "facing", 46 cases (28.40%) tended to "avoidance", and 62 cases (38.27%) tended to "yielding". The exercise self-efficacy score was $58.04 \pm 18.79$ points. The risk perception score was $25.57 \pm 8.37$ points. The exercise outcome expectation score was $29.31 \pm 8.67$ points. The exercise intention score was $12.80 \pm 4.03$ points. The social support score is $32.02 \pm 13.16$ points.

### Comparison of kinesiophobia scores in tumor patients with PICC catheterization with different characteristics is shown in Table 1

TSK-11 scores were significantly associated with several factors, including education level, place of residence, PICC insertion duration, history of complications, pain, and medical coping styles.

### Correlation analysis between kinesiophobia and pain, exercise self-efficacy, risk perception, outcome expectation, exercise intention, social support in tumor patients with PICC

The Pearson correlation analysis results showed that kinesiophobia was positively correlated with pain ($r = 0426$, $P < 0.01$), positively correlated with risk perception ($r = 0.684$, $P < 0.01$), negatively correlated with exercise self-efficacy ($r = -0.677$, $P < 0.01$), negatively correlated with outcome expectation ($r = -0.550$, $P < 0.01$), negatively correlated with exercise intention ($r = -0.524$, $P < 0.01$), and negatively correlated with social support ($r = -0.511$, $P < 0.01$). See Table 2 for details.

### Multiple linear regression analysis of kinesiophobia in tumor patients with PICC

Multiple linear regression was performed with the TSK-11 score of tumor patients with PICC as the dependent variable and the variables with statistically significant differences in univariate analysis as the independent variables ($\alpha_{in} = 0.05$, $\alpha_{out} = 0.10$). The assignment method for categorical variables is shown in Table 3, while the remaining variables are inputted with their original values. The results of multiple linear regression analysis showed that pain, exercise self-efficacy, risk perception level, and catheter insertion duration were factors affecting kinesiophobia in patients with PICC, as shown in Table 4.

## Discussion

### Patients with PICC have a higher incidence of kinesiophobia, but the severity is relatively mild

This study investigated kinesiophobia in a specific population of tumor patients with PICC. The key findings indicate a clinically significant prevalence of kinesiophobia (42.59%), albeit of mild overall severity($20.11 \pm 6.94$), which is shaped by a constellation of factors unique to the cancer journey and the presence of a vascular access device. Notably, the observed prevalence is elevated compared to a report on breast cancer patients from Turkey(30.8%) [29], and it parallels a study on kinesiophobia conducted among postoperative breast cancer patients in Poland(40.8%−42.8%) [30]. This discrepancy

**Table 1. Comparison of Kinesiophobia Scores in Tumor Patients With PICC Catheterization With Different Characteristics (x±s).**

| Item | Number of cases | TSK score | t/F | P |
|---|---|---|---|---|
| Age (years old) | | | 0.388 | 0.699 |
| 18~60 | 53 | 20.12±6.12 | | |
| > 60 | 109 | 19.96±7.33 | | |
| Gender | | | −1.515 | 0.132 |
| Male | 77 | 19.25±7.14 | | |
| Female | 85 | 20.89±6.70 | | |
| Degree of education | | | 7.606 | 0.001 |
| Elementary school and below | 68 | 22.51±7.31 | | |
| Junior and senior high school | 59 | 18.34±6.34 | | |
| University and above | 35 | 18.43±5.87 | | |
| Monthly household income | | | 0.125 | 0.882 |
| ≤5000 | 40 | 20.58±7.87 | | |
| 5000-10000 | 54 | 19.87±6.46 | | |
| > 10000 | 68 | 20.03±6.82 | | |
| Place of residence | | | −2.300 | 0.023 |
| Town | 83 | 18.90±7.33 | | |
| Rural area | 79 | 21.38±6.31 | | |
| Placement duration | | | 9.260 | < 0.001 |
| ≤1 month | 37 | 25.03±8.86 | | |
| 1–3 months | 41 | 18.73±6.43 | | |
| 3–6 months | 41 | 18.71±5.26 | | |
| > 6 months | 43 | 18.53±4.92 | | |
| Catheter location | | | 3.024 | 0.051 |
| Left upper arm | 64 | 21.75±7.83 | | |
| Right upper arm | 94 | 19.04±6.03 | | |
| Other parts | 4 | 19.00±8.72 | | |
| Complications occurrence | | | 7.001 | < 0.001 |
| Yes | 23 | 28.35±5.18 | | |
| No | 139 | 18.75±6.23 | | |
| Foreign body sensation | | | 5.319 | < 0.001 |
| Yes | 87 | 22.57±6.91 | | |
| No | 75 | 17.25±5.83 | | |
| Pain | | | 6.679 | < 0.001 |
| Yes | 26 | 27.50±8.08 | | |
| No | 136 | 18.70±5.73 | | |
| Number of intubation attempts | | | −1.053 | 0.304 |
| ≤1 time | 143 | 19.86±6.70 | | |
| > 1 time | 19 | 22.00±8.52 | | |
| Medical coping modes | | | 5.816 | 0.004 |
| Yielding | 62 | 19.56±7.45 | | |
| Avoidance | 46 | 22.87±6.28 | | |
| Facing | 54 | 18.39±6.24 | | |

**Table 2. Correlation Between TSK-11 Scores of Patients With PICC Catheterization and Various Variables (N=162).**

| Items | TSK-11 | ESE | Risk perception | Outcome expectation | SSRS | Pain score | Exercise intention |
|---|---|---|---|---|---|---|---|
| TSK-11 | 1 | | | | | | |
| ESE | −.677** | 1 | | | | | |
| Risk perception | .684** | −.662** | 1 | | | | |
| Outcome expectation | −.550** | .525** | −.767** | 1 | | | |
| SSRS | −.511** | .470** | −.746** | .655** | 1 | | |
| Pain score | .426** | −.294** | .244** | −.187* | −0.15 | 1 | |
| Exercise intention | −.524** | .485** | −.688** | .606** | .593** | −.213** | 1 |

**When $P<0.01$, the correlation is significant;* when $P<0.05$, the correlation is significant.

**Table 3. Categorical Variable Assignment Method.**

| Independent variables | Assignment |
|---|---|
| Degree of education | Elementary school and below=1; Junior and senior high school=2; University and above=3 |
| Place of residence | Town=1; Rural area=2 |
| Placement duration | ≤1 month=1; 1–3 months=2; 3–6 months=3; > 6 months=4 |
| Complication | No=0, Yes=1 |
| Foreign body sensation | No=0, Yes=1 |
| Coping modes | Yielding=1; Avoidance=2; Facing=3 |
| Pain | No=0, Yes=1 |

**Table 4. Results of Multiple Linear Regression Analysis on Factors Influencing Kinesiophobia in Tumor Patients with PICC (N=162).**

| Independent variable | Regression coefficient | Standard error | Standardized regression coefficient β | t-value | P-value |
|---|---|---|---|---|---|
| Constant | 20.122 | 5.388 | – | 3.735 | < 0.001 |
| Pain | 3.065 | 1.156 | 0.163 | 2.652 | 0.009 |
| PICC carrying duration | −1.248 | 0.356 | −0.200 | −3.506 | 0.001 |
| Risk perception level | 0.279 | 0.088 | 0.336 | 3.157 | 0.002 |
| ESE | −0.096 | 0.025 | −0.261 | −3.891 | < 0.001 |

Note: $R^2=0.674$; $R^2=0.645$; $F=23.504$; $P < 0.001$

with the Turkish study may be partly attributable to differences in sample characteristics. Our study encompassed a heterogeneous mix of solid tumor patients at various disease stages, whereas the cited study focused specifically on breast cancer survivors, a population that may have distinct rehabilitation experiences and support systems. Despite the differing nature of their medical interventions (a retained catheter versus a surgical wound), both groups harbor a fundamental fear of causing internal damage through movement. This fear is potentiated by common experiences of pain, underlying cancer-related fatigue and psychological distress, and maladaptive cognitive appraisals. Specifically, both groups likely exhibit heightened risk perception regarding activity and diminished exercise self-efficacy, as conceptualized within the HAPA model.

The identified prevalence of kinesiophobia situates this phenomenon as a considerable nursing concern in oncology PICC care. Patients are not only managing concerns about catheter-related complications (e.g., displacement, thrombosis) but are also navigating the pervasive effects of their disease and its treatments, such as physical weakness, pain, and

psychological distress [31]. This finding immediately underscores a critical nursing responsibility: to proactively integrate kinesiophobia assessment into routine PICC management. Early identification of misconceptions about movement—such as the irrational belief that normal activity will cause catheter dysfunction—allows nurses to provide timely, corrective education through diverse channels (e.g., demonstrations, graphic manuals) to prevent the entrenchment of avoidant behaviors.

### Synthesis of Influencing Factors and Nursing Implications

Our multivariate analysis revealed a network of modifiable factors, offering a clear roadmap for evidence-based nursing interventions. These factors should not be viewed in isolation but as interconnected targets for a phased management approach.

**The key role of pain and risk perception in early catheterization.** Pain is a significant predictor of kinesiophobia ($\beta = 0.163$, $P = 0.009$), which is consistent with the core viewpoint of the fear avoidance model [32,33] and the findings of Wang [34]. In this study, patients with pain had significantly higher TSK-11 scores than those without pain ($27.50 \pm 8.08$ vs. $18.70 \pm 5.73$). This result can be explained by a dual mechanism: pain fuels the anticipation of movement-induced pain, which in turn activates psychological avoidance and reduces exercise motivation. For the oncology patient, pain sources are multifactorial [35] (e.g., catheter stimulation, tumor infiltration, neuropathies), necessitating a sophisticated nursing response. It is worth noting that in our sample, which included patients with diverse cancer types (e.g., gastrointestinal, lung) and treatment regimens, the experience of pain would be inherently varied. Future research with larger samples could stratify by cancer diagnosis to determine if specific populations, such as those with bone metastases who are at higher risk of pathological fractures [36,37], exhibit disproportionately higher levels of kinesiophobia.

Similarly, an elevated perception of risk regarding catheter-related complications and associated economic burdens was a powerful predictor($\beta = 0.336$, $P = 0.002$), which is consistent with the hypothesis of "risk perception driving health behavior decision-making" in the HAPA theory. This aligns with qualitative work in cancer populations showing that patients often overestimate low-probability risks, leading to unnecessary activity restriction [8,38].

These two factors are paramount in the early phase following PICC insertion. Nursing interventions must be preemptive and multifaceted: Nurses should collaborate with the multidisciplinary team to characterize pain and implement combined strategies, such as pharmacologic analgesia complemented by non-pharmacological interventions (e.g., relaxation techniques, guided imagery). To correct cognitive biases, nurses should use visual aids (e.g., catheter models) and clear, culturally sensitive educational materials to provide a realistic understanding of catheter safety, thereby dismantling fears rooted in the "unknown."

**The protective progression: catheterization duration and self-efficacy.** The negative correlation between catheterization duration and kinesiophobia($\beta = -0.200$, $P = 0.001$) highlights a natural process of adaptation. Patients with catheters in place for over one month exhibited significantly lower kinesiophobia, supporting theories of HAPA where successful lived experience builds confidence [39,40]. Over time, patients accumulate evidence that safe movement is possible, transitioning from intention to sustained action.

This adaptation is critically underpinned by ESE, which we identified as a key protective factor($\beta = -0.261$, $P < 0.001$). Our finding that lower ESE predicts higher kinesiophobia confirms that an individual's belief in their capability to exercise safely is a potent regulator of their behavioral intentions.

These findings define the priorities for the later phase of PICC care. To foster adaptation, nurses should initiate a graduated exercise plan post-insertion, progressing patients from low-intensity walking to supervised upper-limb activities. Concurrently, interventions must explicitly target Exercise Self-Efficacy through evidence-based strategies. These include breaking down activities into manageable, progressive steps (e.g., beginning with 5-minute walks and gradually increasing duration), fostering vicarious learning by facilitating connections with peers who have successfully adapted to PICC, and

using exercise diaries to provide tangible feedback and reinforce mastery. Furthermore, the synergistic effect of social support should be leveraged by actively involving family members in exercise plans [41].

### Research advantages, limitations, and prospects

This study offers several key strengths. Primarily, it introduces the Health Action Process Approach (HAPA) model as a novel theoretical lens for understanding kinesiophobia in PICC patients, thereby providing a cohesive framework for previously disparate factors and addressing a significant gap in the literature. Furthermore, the multicenter design involving three distinct hospitals enhances the generalizability and representativeness of our findings. This study has limitations. Its cross-sectional design prevents causal inference. The relatively small sample size is another limitation of this study. The sample was from a single region, potentially limiting generalizability, and we did not account for variables such as cancer type, disease duration, or prior cancer knowledge, which may influence kinesiophobia. Future research should employ longitudinal designs to track fear dynamics from insertion onward, include more diverse oncology populations, and explore the role of biomarkers. From a practical standpoint, our findings strongly support the development and testing of a staged nursing intervention protocol: prioritizing pain and risk perception management early, and systematically strengthening self-efficacy and social support in the later phases, to comprehensively improve patient quality of life.

## Conclusion

This study shows that the incidence of kinesiophobia in tumor patients with PICC is relatively high, but the degree is mild. Kinesiophobia in tumor patients with PICC is influenced by multiple factors such as pain, duration of catheter placement, risk perception, and exercise self-efficacy. Clinical nursing needs to combine individualized assessment and theory-driven intervention to promote safe patient participation in exercise through pain management, cognitive remodeling, and efficacy enhancement. Further exploration of multidimensional intervention strategies is needed in the future to achieve a behavioral shift from "fear avoidance" to "active adaptation", ultimately improving the long-term prognosis and quality of life of tumor patients. To translate these findings into practice, we propose three concrete steps for nursing care:1.Implement Routine Screening: Integrate the TSK-11 or a brief screener into PICC maintenance visits to systematically identify patients with kinesiophobia. 2. Adopt a Phased Intervention Model: Early phase (≤1 month): Prioritize pain management and risk perception correction through standardized visual education and multidisciplinary analgesia. Later phase (>1 month): Focus on building exercise self-efficacy via graded goal-setting (e.g., progressive walking plans), peer support, and activity diaries. 3.Equip Frontline Nurses: Develop concise, evidence-based training tools (e.g., brief videos, checklists) to enable nurses to deliver these interventions efficiently in routine practice.

## Acknowledgments

The authors are grateful to the patients for their participation. We also acknowledge the reviewers and editors for their conscientious, responsible, expertise and thoughtful feedback, which greatly enhanced this paper. We especially thank Professor Arzu Nurdaş for the meticulous review and valuable insights. Furthermore, we extend our appreciation to Dr. Wang Guoyu for his expert guidance on statistical analysis.

## Author contributions

**Conceptualization:** Yan Wu, Rong Li, Xiaozhu Qiao.

**Data curation:** Xiaohua Zhu, Rong Li.

**Investigation:** Xiaohua Zhu, Yan Wu, Xiaozhu Qiao.

**Methodology:** Rong Li.

**Project administration:** Rong Li.

**Resources:** Ying Yang, Fang Chen.

**Software:** Xiaohua Zhu, Rong Li.

**Supervision:** Ying Yang, Fang Chen.

**Validation:** Xiaohua Zhu.

**Writing – original draft:** Rong Li.

**Writing – review & editing:** Yan Wu, Xiaozhu Qiao.

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
