## [Decision Letter · Decision Letter 0]

13 Oct 2025

PLOS ONE

Dear Dr. Li,

Thank you for submitting your manuscript to PLOS ONE. After careful consideration, we feel that it has merit but does not fully meet PLOS ONE’s publication criteria as it currently stands. Therefore, we invite you to submit a revised version of the manuscript that addresses the points raised during the review process.

The manuscript has been evaluated by two reviewers, and their comments are available below.

The reviewers have raised a number of concerns that need attention. In particular, they request additional information on the methodological and statistical aspects of the study.

Could you please revise the manuscript to carefully address the concerns raised?

We look forward to receiving your revised manuscript.

Kind regards,

Helen Howard

Staff Editor

PLOS ONE

Journal Requirements:

4. In the online submission form, you indicated that data Availability Statement

According to reasonable requirements, the corresponding author Li Rong can provide the dataset used and/or analyzed in this study. Email: lirong19900916@163.com

This study was supported by Taizhou "Fengcheng talents" young scientific and technological talents promotion project (2024). Funding bodies did not play any role in this study.

This study was supported by Taizhou "Fengcheng talents" young scientific and technological talents promotion project (2024). Funding bodies did not play any role in this study.

7. Please update your submission to use the PLOS LaTeX template. The template and more information on our requirements for LaTeX submissions can be found at http://journals.plos.org/plosone/s/latex.

8. We note that there is identifying data in the Supporting Information file <Basic data and analysis.zip>. Due to the inclusion of these potentially identifying data, we have removed this file from your file inventory. Prior to sharing human research participant data, authors should consult with an ethics committee to ensure data are shared in accordance with participant consent and all applicable local laws.

-Location data

Reviewers' comments:

Reviewer's Responses to Questions

**Comments to the Author**

1. Is the manuscript technically sound, and do the data support the conclusions?

Reviewer #1: Partly

Reviewer #2: Yes

2. Has the statistical analysis been performed appropriately and rigorously?

Reviewer #1: Yes

Reviewer #2: Yes

3. Have the authors made all data underlying the findings in their manuscript fully available?

Reviewer #1: No

Reviewer #2: No

4. Is the manuscript presented in an intelligible fashion and written in standard English?

Reviewer #1: Yes

Reviewer #2: Yes

Reviewer #1: Title:

Do not use abbreviations in the title. Please revise the title accordingly.

Abstract:

Introduction: Remove the first and second sentences as it distracts from the main focus of the study. Instead, you may briefly define the catheter or explain the reason for its use.

Results: Report the significance levels (e.g., p, r) appropriately and clearly indicate which findings are statistically significant.

Conclusion: Do not repeat statements that belong in the results section. The conclusions should be clear, concise, and supported by clinical implications.

Keywords: Where are your keywords?

Introduction:

Review and revise according to proper academic writing conventions. The use of headings is currently inappropriate.

Add a reference for the sentence beginning with: “Previous studies on tumor patients with PICC have mostly focused on catheter-related complications...”

In the final paragraph, for the sentence:

“Previous theoretical models related to kinesiophobia often used fear avoidance models, which centered around pain. Nowadays, with the expansion of research subjects on kinesiophobia...”,

include the following reference:

Physical activity, fatigue, kinesiophobia and quality of life: comparative study of prostate cancer survivors with healthy controls. 2025;15:213-220. https://doi.org/10.1136/spcare-2024-005239

Methods:

Please provide a proper reference for the Kendall method and explain the rationale for its use in your study.

Is the Kendall method a statistical analysis method or a method specific to osteoarthritis? Based on this, how were you able to determine your sample?

Why was no power analysis conducted? Is the sample size sufficient? Consider including a post-hoc power analysis.

Justify your use of logistic regression in the statistical analysis section.

Please place the ethics section after the Acknowledgments section, rather than within the main text of the manuscript.

Results:

Were there really no patient dropouts? Please report whether any participants were excluded and provide reasons for exclusions based on the criteria. This information should be presented at the beginning of the results section.

Include a flowchart showing the inclusion and exclusion of participants.

Improve the clarity of the tables and narrative. Emphasize key findings in the text for better understanding.

Discussion:

Numerous cancer-related studies have investigated kinesiophobia. Why did you choose to include literature on coronary and orthopedic conditions instead? Please revise these sections and use more relevant references.

If you choose to divide the discussion into subsections, you must support each section with more comprehensive literature. However, while this format may provide clarity for the methods, it disrupts the continuity—particularly in the paragraph discussing regression results. Consider revising your discussion style.

The studies you cite lack information on sample characteristics. Reporting findings without clarifying the patient populations involved is confusing and disrupts the coherence of the discussion. Please ensure contextual consistency.

Reviewer #2: On a general assessment, the topic under discussion proves to be noteworthy from multiple perspectives. The emerging data and core arguments possess the potential to fill a significant gap in the current literature. However, for the work to achieve its full impact, a deeper analysis of certain key aspects and a clearer establishment of its methodological framework are essential. In summary, this study has a strong foundation and, with careful revision, can leave a more robust and lasting impression in the academic field.

**Do you want your identity to be public for this peer review?** For information about this choice, including consent withdrawal, please see our Privacy Policy

Reviewer #1: No

Reviewer #2: No

---

## [Author Response · Author response to Decision Letter 1]

12 Nov 2025

Dear Editor and Reviewers,

Thank you very much for giving us opportunities to revise our manuscript, and we appreciate the reviewers and editor a lot for your positive and constructive comments and suggestions. We have studied your comments carefully and have made revisions which are marked in red in the “Revised Manuscript With Track Changes”. We hope the corrections will meet with your approval.

Reviewer 1

1 Abstract:

1.1 Introduction: Remove the first and second sentences as it distracts from the main focus of the study. Instead, you may briefly define the catheter or explain the reason for its use.

Response: Thank you for your pertinent suggestions. We have deleted the first sentence and added the background of PICC catheter usage as suggested. We added: “Peripherally Inserted Central Catheters (PICCs) are widely utilized in tumor patients due to their lower risk of complications, extended indwelling duration, reduced local tissue trauma, and overall cost-effectiveness.”

1.2 Results: Report the significance levels (e.g., p, r) appropriately and clearly indicate which findings are statistically significant.

Response: Thank you for your reminder. We have added the values of r and P in the results section. Should our interpretation be inaccurate, we would be grateful for further clarification.

1.3 Conclusion: Do not repeat statements that belong in the results section. The conclusions should be clear, concise, and supported by clinical implications.

Response: We are grateful for the reviewer's valuable input. The manuscript has been modified based on this feedback, and we would be grateful for any further advice should further refinement be needed. The result of our modification was as follows: “The findings underscore a significant clinical concern regarding kinesiophobia among tumor patients with PICC. It is essential for nursing staff to implement systematic assessments and tailored interventions aimed at mitigating kinesiophobia. Addressing this issue can contribute to reducing associated adverse reactions and improving patient mobility and overall quality of life.”

1.4 Keywords: Where are your keywords?

Response: Very sorry, we forgot the keywords due to negligence. Thank you very much for your reminder. The keywords are: Peripherally Inserted Central Catheter; Tumor; Kinesiophobia; Cross-Sectional Study; Exercise Self-efficacy; Health Action Process Approach theoretical.

2 Introduction:

2.1 Review and revise according to proper academic writing conventions. The use of headings is currently inappropriate.

Response: Thanks a lot. We have also made modifications to the beginning and made detailed changes to the "Introduction" section again. We have reorganized the structure and refined the sentence structure, hoping to fully comply with academic writing norms and requirements.

2.2 Add a reference for the sentence beginning with: “Previous studies on tumor patients with PICC have mostly focused on catheter-related complications...”

Response: Thanks a lot. It is indeed necessary to add references here, and we have done so.

2.3 In the final paragraph, for the sentence:

“Previous theoretical models related to kinesiophobia often used fear avoidance models, which centered around pain. Nowadays, with the expansion of research subjects on kinesiophobia...”,

include the following reference:

Physical activity, fatigue, kinesiophobia and quality of life: comparative study of prostate cancer survivors with healthy controls. 2025;15:213-220. https://doi.org/10.1136/spcare-2024-005239

Response: We appreciate the reviewer's valuable suggestion. Accordingly, we have incorporated the recommended reference into the relevant section.

3 Methods:

3.1 Please provide a proper reference for the Kendall method and explain the rationale for its use in your study.

Response: We sincerely appreciate the reviewer's valuable feedback regarding our sample size estimation. We have now included references for the use of the Kendall method. While the rule of using five times the number of variables is an empirical approach, it is generally considered acceptable in cases where well-established scales with clear factor structures are used and when high data quality with good participant compliance is expected. We fully acknowledge the reviewer's valid concern that this method lacks statistical rigor compared to more formal power-based calculations. We will be sure to adopt a more systematic and rigorous sample size calculation in our future studies. Thank you again for this constructive suggestion.

3.2 Is the Kendall method a statistical analysis method or a method specific to osteoarthritis? Based on this, how were you able to determine your sample?

Response: Thank you for your guidance. Kendall is a widely used sample size estimation method at home and abroad. The Kendall method we use is a statistical approach. We calculated based on 5 times the independent variable. There are 11 independent variables in the general data, TSK-11 has 3 dimensions, MCMQ has 3 dimensions, ESE has 1 dimension, Risk perception questionnaire has 3 dimensions, Exercise Results Questionnaire has 1 dimension, Exercise Intention has 2 dimensions, SSRS has 3 dimensions, a total of 27 analytical variables. Five times that is 135 cases.

3.3 Why was no power analysis conducted? Is the sample size sufficient? Consider including a post-hoc power analysis.

Response: We sincerely thank the reviewer for raising this important methodological point. A post-hoc power analysis based on the observed effect sizes indicated that a total sample of 323 would be ideal. Our final sample of 162 participants was initially determined by a widely cited empirical rule of thumb in similar clinical survey studies, which suggests a target of 5-10 participants per predictor variable. While we fully acknowledge that the achieved sample size is lower than the post-hoc calculation suggests, we would be grateful if the reviewer could consider that this empirical approach, while less precise than an a priori power analysis, is a recognized and pragmatic method in exploratory clinical research, particularly under constraints of time and patient accessibility. Importantly, despite the smaller sample, the key predictors in our regression model demonstrated substantial and statistically significant effects. We have transparently acknowledged the sample size limitation in the revised manuscript and recommend future validation with larger, prospectively powered studies.

Z2=1.96*1.96=3.8416; p=0.30; δ=0.05

N=323

[1]Gencay Can A, Can SS, Ekşioğlu E, Çakcı FA. Is kinesiophobia associated with lymphedema, upper extremity function, and psychological morbidity in breast cancer survivors?. Turk J Phys Med Rehabil. 2018;65(2):139-146.

3.4 Justify your use of logistic regression in the statistical analysis section.

Response: We appreciate the reviewer for pointing out this discrepancy. The analysis has been correctly conducted using multiple linear regression, not binary logistic regression. We have revised the manuscript accordingly to ensure the statistical methods are accurately described throughout.

3.5 Please place the ethics section after the Acknowledgments section, rather than within the main text of the manuscript.

Response: Thank you for the opportunity to clarify. We note that the ethical statement is presented within the main text of many recent publications in your journal. Following this precedent, we included it in the main body of our manuscript. We sincerely apologize if this does not align with the specific requirements for our submission and humbly submit that we can move it to another section immediately upon request.

4 Results:

4.1 Were there really no patient dropouts? Please report whether any participants were excluded and provide reasons for exclusions based on the criteria. This information should be presented at the beginning of the results section.

Response: Based on our analysis, there were indeed no patients who withdrew midway. It can be attributed to the following factors: First, the fixed weekly schedule for PICC maintenance provided participants with repeated and flexible opportunities to complete the survey during their routine clinic visits, thereby minimizing the risk of incomplete responses. Second, prior to the study, the research team dedicated substantial effort to standardizing survey procedures and communicating with patients in clear, accessible language. This ensured that participants could provide fully informed consent based on a thorough understanding of the study. Finally, the consistent and familiar environment of the vascular access clinic, along with the established trust between patients and the regular nursing staff, contributed to a supportive and cooperative atmosphere, facilitating continued participation. (Thank you for your suggestion. We have provided a brief analysis at the beginning of the results section.)

We have excluded three patients with stroke combined with tumor disease due to their own movement disorders.

4.2 Include a flowchart showing the inclusion and exclusion of participants.

Response: Thank you sincerely for your suggestion. Based on the inclusion and exclusion criteria, we have presented the following flowchart. If it is not feasible, we kindly request further guidance

4.3 Improve the clarity of the tables and narrative. Emphasize key findings in the text for better understanding.

Response: We are grateful for the valuable suggestion. Based on this comment, we have revised the relevant paragraphs and incorporated the key findings from the univariate analysis to strengthen the manuscript. We would appreciate any further clarification should additional modifications be necessary.

5 Discusion

5.1 Numerous cancer-related studies have investigated kinesiophobia. Why did you choose to include literature on coronary and orthopedic conditions instead? Please revise these sections and use more relevant references.

Response: We thank the reviewer for this insightful comment. We agree that comparing our findings to studies on other cancer populations is more appropriate. In response, we have revised the discussion by removing comparisons with coronary and orthopedic conditions and have instead incorporated relevant references from oncology literature to provide a more focused context for our findings.

5.2 If you choose to divide the discussion into subsections, you must support each section with more comprehensive literature. However, while this format may provide clarity for the methods, it disrupts the continuity—particularly in the paragraph discussing regression results. Consider revising your discussion style.

Response: We sincerely appreciate the reviewer's insightful observation on the structure of our discussion. The discussion has been thoroughly revised in response, with the aim of enhancing its coherence and depth. We would be thankful for any further guidance the reviewer might offer, should any part require further refinement.

5.3 The studies you cite lack information on sample characteristics. Reporting findings without clarifying the patient populations involved is confusing and disrupts the coherence of the discussion. Please ensure contextual consistency.

Response: We thank the reviewer for this important observation. We have now revised the discussion to include relevant details (e.g., incidence of kinesiophobia) for the cited references to ensure a more coherent and meaningful comparison with our findings. And we further searched for relevant literature and read it thoroughly, adding comparisons and references to some of the literature.

For example, adding the following content to the discussion section

Notably, the observed prevalence is elevated compared to a report on breast cancer patients from Turkey (30.8%) [30], and it parallels a study on kinesiophobia conducted among postoperative breast cancer patients in Poland (40.8%-42.8%) [31]. This discrepancy with the Turkish study may be partly attributable to differences in sample characteristics. Our study encompassed a heterogeneous mix of solid tumor patients at various disease stages, whereas the cited study focused specifically on breast cancer survivors, a population that may have distinct rehabilitation experiences and support systems.

For the oncology patient, pain sources are multifactorial [36] (e.g., catheter stimulation, tumor infiltration, neuropathies), necessitating a sophisticated nursing response. It is worth noting that in our sample, which included patients with diverse cancer types (e.g., gastrointestinal, lung) and treatment regimens, the experience of pain would be inherently varied. Future research with larger samples could stratify by cancer diagnosis to determine if specific populations, such as those with bone metastases who are at higher risk of pathological fractures [37-38], exhibit disproportionately higher levels of kinesiophobia.

Reviewer 2

On a general assessment, the topic under discussion proves to be noteworthy from multiple perspectives. The emerging data and core arguments possess the potential to fill a significant gap in the current literature. However, for the work to achieve its full impact, a deeper analysis of certain key aspects and a clearer establishment of its methodological framework are essential. In summary, this study has a strong foundation and, with careful revision, can leave a more robust and lasting impression in the academic field.

Response: We are truly grateful to the reviewer for their generous comments regarding the strong foundation of our study and its potential impact. We also sincerely appreciate the insightful guidance on how to enhance the work. The recommendation to perform a deeper analysis and to more clearly establish the methodological framework is well-received. Our team will diligently undertake the careful revision suggested to improve the robustness and clarity of the manuscript.

1 A literature review is present, but the gap is not clearly specified.

Response: We thank the reviewer for this critical observation. We agree that a clearly defined research gap is essential. In response, we have thoroughly revised the introduction section, try to explicitly state the research gap. We have now clearly articulated that while kinesiophobia has been studied in other patient groups, there is a lack of focused research on tumor patients with PICC, especially one that employs a theoretical framework to investigate the influencing factors. The modifications can be found on page 4-5 of the revised manuscript.

If the changes are not suitable, we kindly request further clarification from you. Thank you very much.

2 How HAPA theory was integrated into the study (e.g., variable selection) is not explained.

Response: Thank you for your pieces of advice. The selection of the HAPA model is justified in the introduction's fourth paragraph. We consider kinesiophobia to be a specific stage, dynamic and unhealthy behavior. This model is particularly suited to our research context as it effectively explains the progression from motivational intentions to volitional action in health behaviors. A key rationale for its adoption is its emphasis on the volitional, or actional, phase of change—a process critically mediated by factors such as self-efficacy, coping strategies, and action planning. Consequently, the model directly informed our choice of measurement constructs, thus ensuring our variable selection is theoretically grounded. Guided by this model, we have selected key constructs for measurement, including self-efficacy, risk perception, outcome expectations, and behavioral intention.

3 A clear research hypothesis should be added.

Response: Thank you for your pertinent suggestions. We have referred to previous publications in this journal and it seems that there is no specific requirement to write a hypothesis section. So, it is indeed our negligence in this regard. Can we make the assumption as follow: ①The study hypothesized that key constructs from the HAPA model, including lower exercise self-efficacy, higher risk perception, and lower exercise intention, would be significantly associated with higher levels of kinesiophobia; ②We hypothesized that a longer PICC indw

---

## [Decision Letter · Decision Letter 1]

15 Dec 2025

Dear Dr. Li,

Two reviewers have evaluated the manuscript. While one reviewer accepts the manuscript in its current version, the other has made some minor suggestions for improvement before it can be accepted for publication.

We look forward to receiving your revised manuscript.

Kind regards,

Christoph Strumann

Academic Editor

PLOS One

Journal Requirements:

Reviewers' comments:

Reviewer's Responses to Questions

**Comments to the Author**

Reviewer #1: (No Response)

Reviewer #3: (No Response)

2. Is the manuscript technically sound, and do the data support the conclusions?

Reviewer #1: Yes

Reviewer #3: Yes

3. Has the statistical analysis been performed appropriately and rigorously?

Reviewer #1: Yes

Reviewer #3: Yes

4. Have the authors made all data underlying the findings in their manuscript fully available?

Reviewer #1: Yes

Reviewer #3: Yes

5. Is the manuscript presented in an intelligible fashion and written in standard English?

Reviewer #1: Yes

Reviewer #3: Yes

Reviewer #1: Dear Authors,

I commend your diligent efforts and comprehensive responses to every reviewer comment on the revised manuscript (PONE-D-25-33789R1). As a result of these revisions, both the methodological rigor and clinical validity of the study have been significantly strengthened. Our concerns regarding methodological transparency—specifically the validation of multiple linear regression assumptions , the justification of variable selection within the HAPA framework , and the candid discussion of the sample size limitation —have been fully addressed. The greatest strength of this work lies in its focus on a crucial nursing concern, kinesiophobia in oncology patients with a PICC , and its use of the HAPA model to provide a clear roadmap for clinically applicable, phased interventions (early pain/risk management; later self-efficacy enhancement). The revised manuscript now meets publication standards and successfully fills a significant gap in the existing literature. I recommend its acceptance in its current form.

Reviewer #3: I assessed its improved state with the previous revision. I have added a few minor revision notes to the work. It was suggested that the impact of working in nursing care should be assessed and that concrete recommendations should be developed.

**Do you want your identity to be public for this peer review?** For information about this choice, including consent withdrawal, please see our Privacy Policy

Reviewer #1: No

Reviewer #3: **Yes:** Asist Prof Dr Arzu Nurdaş

---

## [Author Response · Author response to Decision Letter 2]

17 Dec 2025

Dear Reviewers,

We sincerely thank the editor and reviewers for the opportunity to revise our manuscript and for their valuable, constructive feedback. We have carefully considered all comments and are pleased to resubmit a revised version. All modifications made in response to the feedback are highlighted in the manuscript with track changes. We hope our revisions are now satisfactory and that the manuscript is much improved for publication.

1.This sentence is getting off topic. ”Chemotherapy is required in approximately 72.6% of cancer cases”

Response: Thanks for pointing out that the sentence regarding the prevalence of chemotherapy was diverting from the core narrative. Upon reviewing, we indeed identified a problem of logical discontinuity. We had changed as follow: For patients requiring chemotherapy, the peripherally inserted central catheter (PICC) has become the most widely used central venous access in China due to its minimally invasive nature, avoidance of repeated punctures, and convenience for safe medication. Should the modifications fall short, we would appreciate your further guidance.

2.What re these conditions and how do they differ from physical activity disorder?

Response: We thank the reviewer for this critical observation, which has helped us clarify a key methodological point. We agree that the original terms "contraindications to exercise" and "physical activity disorders" were ambiguous and potentially overlapping. We have revised the exclusion criteria in the Methods section to precisely differentiate between them:

“Absolute contraindications to physical activity” now refers to specific, medically prohibitive conditions (e.g., unstable cardiovascular disease).

“Severe neurological or musculoskeletal disorders” is used to describe conditions that cause substantial physical limitation. The rationale for this exclusion is to ensure that a high score on the kinesiophobia scale truly reflects a fear of movement rather than an inability to move due to organic pathology, thereby preserving the construct validity of our primary outcome.

Guided by your comment, we believe these clarifications strengthen the methodological rigor of our study. Please see the updated as follows:

①Have absolute contraindications to physical activity (e.g., unstable cardiovascular disease, acute systemic infection, or other conditions where exercise is medically prohibited as determined by the treating physician).

②Have severe comorbid conditions that could independently limit survival or confound the assessment (e.g., severe heart failure [NYHA Class III/IV], end-stage renal disease on dialysis, or severe immunocompromised state).

③Have a documented history of mental illness or cognitive impairment (e.g., dementia, schizophrenia) that would impede the ability to provide informed consent or accurately report symptoms.

④Have severe neurological or musculoskeletal disorders that substantially limit voluntary limb movement (e.g., paralysis, severe Parkinson's disease, or advanced osteoarthritis preventing basic arm or leg movement), making the assessment of kinesiophobia related to fear rather than physical incapacity unreliable.

Looking forward to your further guidance.

3.p<0,001 if bonferoni correction was made it should be stated or found to be highly significant.

Response: We thank the reviewer for raising this important methodological point regarding multiple comparisons. In our univariate analysis (Table 1), the P-values reported for multi-category variables (e.g., education level, placement duration) were derived from the initial omnibus tests (e.g., one-way ANOVA), not from subsequent pairwise comparisons. As our primary aim at this stage was variable screening for the regression model (using a threshold of P < 0.05), and we did not conduct or report post-hoc pairwise comparisons between individual groups, a Bonferroni correction was not required. To prevent any ambiguity, we have revised the ‘Statistical methods’ section to clarify this approach explicitly. We now state: “Variables yielding a P-value < 0.05 in these initial omnibus tests were considered candidates for multivariate analysis. We did not perform post-hoc pairwise comparisons; therefore, a Bonferroni correction was not applied.” We hope this clarification satisfactorily addresses the reviewer's concern.

Should any part require further refinement, we warmly welcome your expert advice and would deeply appreciate your additional feedback.

4.A few concrete suggestions regarding the impact of the study on nursing care could be added to the conclusion or discussion section.

Response: We sincerely thank the reviewer for this constructive suggestion to enhance the practical impact of our study. To maintain conciseness in the Discussion, we have consolidated the concrete, actionable suggestions into the Conclusion section. In direct response, we have revised the Conclusion section to incorporate concrete, actionable recommendations for nursing practice. These are structured around a phased intervention model derived from our findings: We now added:

…ultimately improving the long-term prognosis and quality of life of tumor patients. “To translate these findings into practice, we propose three concrete steps for nursing care:1.Implement Routine Screening: Integrate the TSK-11 or a brief screener into PICC maintenance visits to systematically identify patients with kinesiophobia. 2. Adopt a Phased Intervention Model: Early phase (≤1 month): Prioritize pain management and risk perception correction through standardized visual education and multidisciplinary analgesia. Later phase (>1 month): Focus on building exercise self-efficacy via graded goal-setting (e.g., progressive walking plans), peer support, and activity diaries. 3.Equip Frontline Nurses: Develop concise, evidence-based training tools (e.g., brief videos, checklists) to enable nurses to deliver these interventions efficiently in routine practice.”

We believe these specific additions bridge the gap between research findings and clinical application, directly addressing the reviewer's recommendation.

We greatly appreciate the efficient, professional and rapid processing of our paper by your team. If there is anything else we should do, please do not hesitate to let us know.

Thank you once again for your diligent work and valuable guidance on our manuscript. As the year draws to a close, we hope you have the opportunity to enjoy a restful and festive holiday season. Best wishes for Christmas and a successful New Year.

Sincerely,

Rong Li, Xiaohua Zhu, Yan Wu, Xiaozhu Qiao, Ying Yang, Fang Chen

Email�lirong19900916@163.com

2025.12.16

---

## [Editor Report · Decision Letter 2]

18 Dec 2025

Analysis of the current status and influencing factors of kinesiophobia in tumor patients with Peripherally Inserted Central Catheter� A cross-sectional study

PONE-D-25-33789R2

Dear Dr. Li,

We’re pleased to inform you that your manuscript has been judged scientifically suitable for publication and will be formally accepted for publication once it meets all outstanding technical requirements.

Kind regards,

Christoph Strumann

Academic Editor

PLOS One
---

## [Editor Report · Acceptance letter]

PONE-D-25-33789R2

PLOS One

Dear Dr. Li,

I'm pleased to inform you that your manuscript has been deemed suitable for publication in PLOS One. Congratulations! Your manuscript is now being handed over to our production team.

Kind regards,

on behalf of

Dr. Christoph Strumann

Academic Editor

PLOS One